# Passive and active immunity in infants born to mothers with SARS-CoV-2 infection during pregnancy: prospective cohort study

Dongli Song,[1,2] Mary Prahl,[3] Stephanie L Gaw [ID] ,[4] Sudha Rani Narasimhan,[1,2] Daljeet S Rai,[5] Angela Huang,[1] Claudia V Flores,[1] Christine Y Lin,[4] Unurzul Jigmeddagva,[3] Alan Wu,[6] Lakshmi Warrier,[7] Justine Levan,[7] Catherine B T Nguyen,[7] Perri Callaway,[7] Lila Farrington,[7] Gonzalo R Acevedo,[7] Veronica J Gonzalez,[4] Anna Vaaben,[7] Phuong Nguyen,[8] Elda Atmosfera,[8] Constance Marleau,[8] Christina Anderson,[1,2] Sonya Misra,[1,2] Monica Stemmle,[2,9] Maria Cortes,[1] Jennifer McAuley,[1] Nicole Metz,[1] Rupalee Patel,[1] Matthew Nudelman,[10] Susan Abraham,[2,9] James Byrne,[11,12] Priya Jegatheesan [ID] [1,2]

► Prepublication history and supplemental material for this paper is available online. To view these files, please visit the journal online (http://dx.doi.org/10.1136/bmjopen-2021-053036).

**Correspondence to**
Dr Priya Jegatheesan;
Priya.Jegatheesan@hhs.sccgov.org

## ABSTRACT

**Objective** To investigate maternal immunoglobulins' (IgM, IgG) response to SARS-CoV-2 infection during pregnancy and IgG transplacental transfer, to characterise neonatal antibody response to SARS-CoV-2 infection, and to longitudinally follow actively and passively acquired antibodies in infants.

**Design** A prospective observational study.

**Setting** Public healthcare system in Santa Clara County (California, USA).

**Participants** Women with symptomatic or asymptomatic SARS-CoV-2 infection during pregnancy and their infants were enrolled between 15 April 2020 and 31 March 2021.

**Outcomes** SARS-CoV-2 serology analyses in the cord and maternal blood at delivery and longitudinally in infant blood between birth and 28 weeks of life.

**Results** Of 145 mothers who tested positive for SARS-CoV-2 during pregnancy, 86 had symptomatic infections: 78 with mild-moderate symptoms, and 8 with severe-critical symptoms. The seropositivity rates of the mothers at delivery was 65% (95% CI 0.56% to 0.73%) and the cord blood was 58% (95% CI 0.49% to 0.66%). IgG levels significantly correlated between the maternal and cord blood (Rs=0.93, p<0.0001). IgG transplacental transfer ratio was significantly higher when the first maternal positive PCR was 60–180 days before delivery compared with <60 days (1.2 vs 0.6, p<0.0001). Infant IgG seroreversion rates over follow-up periods of 1–4, 5–12, and 13–28 weeks were 8% (4 of 48), 12% (3 of 25), and 38% (5 of 13), respectively. The IgG seropositivity in the infants was positively related to IgG levels in the cord blood and persisted up to 6 months of age. Two newborns showed seroconversion at 2 weeks of age with high levels of IgM and IgG, including one premature infant with confirmed intrapartum infection.

**Conclusions** Maternal SARS-CoV-2 IgG is efficiently transferred across the placenta when infections occur more than 2 months before delivery. Maternally derived passive immunity may persist in infants up to 6 months of life. Neonates are capable of mounting a strong antibody response to perinatal SARS-CoV-2 infection.

## Strengths and limitations of this study

► This study included pregnant mothers with SARS-CoV-2 infection in all three trimesters of pregnancy and provided a comprehensive understanding of maternal SARS-CoV-2 IgG transplacental transfer throughout pregnancy.

► This is the first longitudinal study that has followed maternally derived SARS-CoV-2 IgG in infants up to 28 weeks.

► This is the first study, to our knowledge, that characterised neonatal serology response to perinatal SARS-CoV-2 in two neonates.

► In asymptomatic mothers who were identified as SARS-CoV-2 PCR positive at the time of delivery, we were unable to ascertain the precise timing of infection.

► The cohort had few severe cases of maternal infection and premature births before 35 weeks of gestation.

## INTRODUCTION

Our understanding of the immune response to SARS-CoV-2 is expanding rapidly through extensive basic and clinical studies.[1–4] However, the literature on SARS-CoV-2 immunity in pregnant mothers and infants remains limited.[5–9] Global efforts are focused on controlling the COVID-19 pandemic through public health prevention measures and vaccination. Knowledge of neonatal immune

response to SARS-CoV-2 and maternally derived passive immunity in young infants is urgently needed to inform ongoing COVID-19 infection prevention and vaccination strategies to protect pregnant mothers and infants.

The physiological changes occurring during pregnancy make the mothers more vulnerable to severe respiratory infections. The Centers for Disease Control (CDC) reported that COVID-19 infection poses a significantly higher risk of severe illness and death in symptomatic infected pregnant than symptomatic infected non-pregnant women.[10] An international study collected outcomes of 706 pregnant mothers with SARS-CoV-2 infection during pregnancy and their newborns from 18 developed and developing countries.[11] Results from this large-scale study demonstrated that pregnant women with COVID-19, compared with those without COVID-19, were at a substantially increased risk of severe pregnancy complications and death. Interestingly, several cohort studies conducted in the USA have found that the majority of pregnant women with SARS-CoV-2 infection were either asymptomatic or had mild symptoms.[5 12–16]

Neonatal infection following birth to a mother with SARS-CoV-2 infection during pregnancy is infrequent.[16–21] CDC reported that the perinatal SARS-CoV-2 infection rate among infants born to mothers with COVID-19 during pregnancy was 2.6%.[17] Notably, the majority of the infected infants were born to the mothers who had infections within 1 week of delivery. A meta-analysis review of 174 neonatal infection cases found that 70% and 30% of infections are due to environmental and vertical transmission, respectively. Fifty-five per cent of infected neonates were symptomatic, including fever (44%), gastrointestinal (36%), respiratory (52%) and neurological manifestations (18%).[22] Data from CDC and other case studies showed that the majority of infected neonates were asymptomatic or exbibit mild symptoms.[16 17 23] However, in a UK national population-based cohort study, 42% of the infected infants presented with severe symptoms.[20 24] The large-scale international investigation found that infants born to women with COVID-19 during pregnancy had a significantly higher risk of severe perinatal mobility and mortality.[11] These risks remained significant after adjusting for prematurity, indicating a direct effect of COVID-19 on the infants.

Children are more vulnerable to severe respiratory infections. Interestingly, children, compared with adults, are less susceptible to SARS-CoV-2 infection and less likely to develop severe illness.[25 26] However, children with certain underlying medical conditions and infants (age <1 year) might be at increased risk of severe illness from SARS-CoV-2 infection.[27 28] A nationwide Chinese study included 2135 children with confirmed and suspected SARS-CoV-2 infection.[28] Among them, 18% were less than 1 year old, and 10% of the infants had severe or critical clinical symptoms. In a small case series from the USA, all 18 less than 3-month-old infants with COVID-19 presented with mild symptoms.[29]

An important aspect of immunity against infectious pathogens in young infants relies on effective maternal antibody production, transfer of maternal antibodies across the placenta to the fetus and persistence of passive immunity in the infant. Recent publications have shown evidence of maternal SARS-CoV-2 antibody transplacental transfer.[6 7 9] However, the majority of maternal SARS-CoV-2 infections in these reports occurred late in pregnancy, as these studies were conducted during the first few months of the COVID-19 pandemic. Therefore, the timing and efficiency of maternal antibody production and transplacental transfer throughout gestation remain to be fully understood, which has important implications for the timing of maternal immunisation to benefit both pregnant mothers and their young infants. Furthermore, the important question as to the persistence of maternally derived passive immunity in infants needs to be investigated. While SARS-CoV-2 infection has been described in newborns,[17 22] little is known about infant immune response to perinatal infection. The aims of this study were to investigate SARS-CoV-2 antibody transplacental transfer with respect to the timing of maternal infection during gestation, antibody response to SARS-CoV-2 infection in the newborns, and persistence of passively and actively acquired SARS-CoV-2 antibodies in infants.

## METHODS
### Study design, participants and procedures
This is a prospective observational study of pregnant mothers with SARS-CoV-2 infection during pregnancy and their infants. The study was conducted from 15 April 2020 to 31 March 2021, in a public healthcare system, including one regional medical centre and two community hospitals. The healthcare system primarily serves the medically indigent population of Santa Clara County California (USA).

On 15 April 2020, our institution implemented universal SARS-CoV-2 screening protocol in labour and delivery units. All women who were admitted for delivery or within 3 days prior to admission for elective deliveries were tested for SARS-CoV-2 by PCR using a nasopharyngeal swab.[30] From October 2020 onwards, women who tested positive within 90 days prior to admission for delivery and did not have new symptoms of COVID-19 were not retested at the time of delivery. In addition to PCR testing, mothers were screened for history of SARS-CoV-2 infection and PCR testing during pregnancy. PCR tests were done anytime during pregnancy if the mother experienced symptoms concerning for COVID-19 or had close contact with a person with COVID-19. The pregnant mothers who had a documented positive SARS-CoV-2 PCR during the current pregnancy, either prior to admission or tested positive after admission, were eligible for the study. We screened and enrolled mothers for the study after they were admitted to the labour and delivery units. The timing of maternal SARS-CoV-2 infection was based on the first positive SARS-CoV-2 PCR test. The severity of

SARS-CoV-2 symptoms (mild, moderate, severe or critical) was assessed according to the Society for Maternal-Fetal Medicine guidelines.[31]

If the maternal infection was within 10–14 days of delivery, the mother and infant roomed in together with airborne isolation precautions and the mother wore a surgical mask when holding and breast feeding the baby during the isolation period. The nasopharyngeal SARS-CoV-2 PCR was performed in the newborns at 24 hours of life. The infants were retested between 48 and 72 hours of life if they were in the neonatal intensive care unit (NICU). Maternal and neonatal nasopharyngeal samples were collected according to hospital standard procedure. PCR tests were performed by hospital clinical laboratories using validated SARS-CoV-2 assays for clinical diagnosis (online supplemental methods).

Maternal and cord blood were collected at the time of delivery. Serial infant blood samples were initially designed to be collected at 2 weeks, 2 months and 6 months coordinated with routine paediatric clinic visits. During the pandemic, the visit schedules varied significantly due to parental hesitance to come to the clinics for concerns of COVID-19 exposure. Thus, infants' blood samples were collected anytime between 1–4 weeks, 5–12 weeks and 13–28 weeks at the time of clinic visits. Levels of SARS-CoV-2 IgM and IgG to the spike protein receptor binding domain and nucleocapsid protein of SARS-CoV-2 were measured using the Pylon 3D automated immunoassay system (ET Healthcare, Palo Alto, California, USA) as previously described.[32] The background corrected signal was reported as relative fluorescent units (RFU), which is proportional to the amount of specific antibodies in the sample allowing for quantification. The positive cut-offs for IgM and IgG were set to >50 RFU to achieve 100% specificity and a high level of sensitivity.[32] Quantitative reverse transcriptase PCR was performed on maternal blood, cord blood, placenta and meconium in a subset of infants. Primer sequences targeted the N and Orf1b SARS-CoV-2 genes (online supplemental methods, online supplemental tables 1 and 2).

### Data collection and analysis
Clinical data included maternal and neonatal demographics, the severity of maternal symptoms of SARS-CoV-2 infection, days between maternal first positive SARS-CoV-2 PCR test and delivery, and neonatal outcomes. Demographics, clinical outcomes, and serum IgM and IgG levels were summarised using descriptive analyses. Transplacental IgG transfer ratios were calculated by dividing cord blood IgG levels by maternal blood IgG levels. Correlation between maternal and cord blood IgG levels and correlation between placental transfer ratio and gestational age (GA) at birth were analysed using Spearman's rank-order correlation. The transfer ratios were compared between maternal groups based on infection severity and time between first maternal positive PCR and delivery using the Kruskal-Wallis test, followed

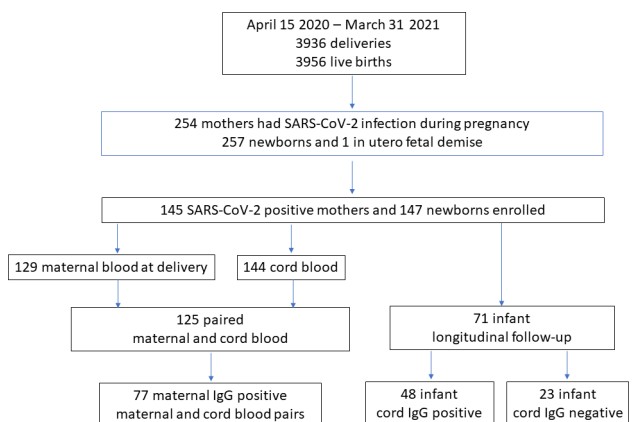

**Figure 1** Study participants' enrolment.

by Dunn's test for pairwise multiple comparisons with the Holm-Sidák stepwise adjustment.

### Patient and public involvement
Patients or the public were not involved in the design, or conduct, or reporting, or dissemination plans of our research.

## RESULTS
During the study period, 3936 mothers delivered in the health system with 3956 live births, and 254 (6.5%) of the mothers had at least one positive SARS-CoV-2 PCR test during the pregnancy. The study enrolled 145 mothers with SARS-CoV-2 infection and 147 of their infants (figure 1). Of 145 enrolled mothers, 86 (59%) had symptomatic infection, including 78 with mild-moderate symptoms and 8 with severe-critical symptoms (table 1). The distribution of the severity of the maternal infection is shown in online supplemental table 3. Of 147 newborns, 23 (16%) were admitted to the NICU. SARS-CoV-2 PCR was performed on nasopharyngeal specimens of 89 newborns at 24 hours of life, and only one 31-week preterm infant tested positive.

### Maternal and cord blood serology
Serum serology was performed on 129 mothers at delivery and 144 cord blood samples. The temporal profiles of maternal blood IgM and IgG with respect to the timing of first maternal PCR positivity are shown in figure 2A,B. Antibody status and levels in maternal and cord blood were evaluated in four groups based on the days between maternal first positive SARS-CoV-2 PCR and delivery (<14 days, 14–59 days, 60–180 days and >180 days) (table 2). The maternal IgG level at the time of delivery was significantly lower in the group aged <14 days compared with 14–59 days, 60–180 days and >180 days (p=0.0001) (figure 2C and table 2). The overall maternal seropositivity rate at delivery was 65% (84 of 129, 95% CI 0.56% to 0.73%). Of the 31 mothers who were asymptomatic and identified positive for SARS-CoV-2 at delivery, 10 had serology tests positive for IgG but negative for IgM, consistent with convalescent infections. The temporal

**Table 1** Maternal and neonatal demographics and outcomes

|  | Maternal and infant serology cohort |
| --- | --- |
| Mothers, n | 145 |
| Newborns, n | 147 |
| Maternal demographics and outcomes | |
| Maternal age, years, median (range) | 27 (16–42) |
| Gravida, median (range)* | 3 (1–12) |
| Para, median (range)† | 1 (0–9) |
| Hispanic, n (%) | 126 (87) |
| Race | |
| White, n (%) | 130 (90) |
| Black, n (%) | 6 (4) |
| Asian, n (%) | 9 (6) |
| Asymptomatic, n (%) | 59 (41) |
| Mild to moderately symptomatic, n (%) | 78 (54) |
| Severe to critically symptomatic, n (%) | 8 (6) |
| Symptomatic at the time of delivery, n (%) | 22 (15) |
| Caesarean section, n (%) | 46 (32) |
| Multiple pregnancies, n (%) | 3 (2) |
| Maternal diabetes, n (%) | 29 (20) |
| Maternal hypertension, n (%) | 30 (21) |
| Maternal obesity, n (%) | 33 (23) |
| Preterm delivery, n (%) | 15 (10) |
| Intrauterine fetal demise, n (%) | 1 (1) |
| Neonatal demographics and outcomes | |
| Gestational age, weeks, median (range) | 39.1 (27.4–41.6) |
| Birth weight, grams, median (range) | 3285 (990–4670) |
| Breast feeding in the hospital, n (%) | 143 (97) |
| Exclusive breast feeding in the hospital, n (%) | 85 (58) |
| Rooming in with mother, n (%) | 132 (90) |
| NICU admission, n (%)‡ | 23 (16) |
| Length of stay during birth hospital, days, median (range) | 2 (1–81) |
| SARS-CoV-2 nasopharyngeal swab positive, n (%)§ | 1 (1) |

*Gravida—number of pregnancies.
†Para—number of deliveries.
‡Reasons for NICU admissions: 7 for prematurity, 1 for a congenital anomaly, 1 for dehydration and 14 for respiratory distress, metabolic acidosis and or evaluation for infection.
§SARS-CoV-2 PCR using nasopharyngeal specimens was performed in 70 (99%) of the newborns born to mothers who were first PCR positive within 2 weeks of delivery.
NICU, neonatal intensive care unit.

profile of cord blood IgG with respect to the timing of first maternal PCR positivity is shown in figure 2D. The cord blood IgG positivity rate was 58% (83 of 144, 95% CI 0.49% to 0.66%).

Paired serology analysis was performed in 125 maternal cord blood samples (table 2). Of the 77 IgG positive mothers, 69 (90%) of their newborns' cord sera were positive for IgG. Of the eight IgG negative infants, seven were born to mothers with infection within 45 days of delivery, and one was born to a mother who had a positive PCR at 254 days before delivery. Of the 48 IgG negative mothers, 45 (94%) of their newborns' cord sera were negative for IgG. Of the 125 cord samples, there were 3 infants whose cord blood was positive for IgM (65, 136 and 62 RFU). Notably, all three were born to mothers whose blood was also positive for IgM at the time of delivery. The follow-up serology tests for two of the infants at 2 and 3 weeks of age were negative for IgM and IgG. No follow-up serology was available for the third infant. Available delivery specimens (maternal and cord blood, placenta and meconium)

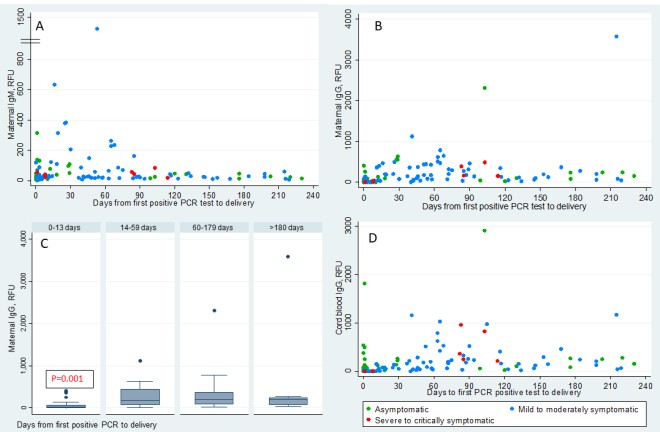

**Figure 2** Temporal distribution of maternal and cord blood IgM and IgG. (A,B) Scatterplots show the distribution of maternal blood SARS-CoV-2 IgM and IgG levels in relative fluorescent unit (RFU) at the time of delivery in y-axis and days from maternal first positive SARS-CoV-2 PCR test to delivery in x-axis. (C) Box plot of the distribution of the maternal IgG levels at the time of delivery in the maternal groups based on the number of days between maternal infection and delivery. The box represents the IQR from 25th to 75th percentile (IQR). The marker within the box is the median and the 'whiskers' reach the 1.5 times IQR. (D) Scatterplots show the distribution of cord blood SARS-CoV-2 IgG levels in RFU at the time of delivery in y-axis and days from maternal first positive SARS-CoV-2 PCR test to delivery in x-axis. The different colours represent the severity of the maternal symptoms at the time of diagnosis.

were evaluated by SARS-CoV-2 PCR and found to be negative for all three infants (online supplemental table 4). Two of these were term infants and had a normal newborn course in the hospital and remained asymptomatic during the first month of life. The third infant was a 31-week GA premature infant who was delivered due to in utero growth restriction. This infant had typical respiratory symptoms due to lung immaturity. The chest X-ray did not show any evidence of infiltration. The infant was on continuous positive airway pressure (CPAP) and nasal cannula, with 21% fractional inspired oxygen (FiO$_2$) for 3 weeks.

There was a significant positive correlation between IgG levels in the 125 paired maternal and cord blood samples (Rs=0.93, p<0.0001, figure 3A). Transplacental IgG transfer ratios were calculated in 77 IgG positive mothers, and the median transfer ratio was 1.0 (95% CI 0.86 to 1.09). The transfer ratio was significantly higher in the mothers who were severe-critically symptomatic (1.6, 95% CI 1.42 to 2.49, n=4) compared with mothers who were asymptomatic (1.0, 95% CI 0.62 to 1.14, n=23) (1.6 vs 1.0, p=0.003) or mild-moderately symptomatic (0.9, 95% CI 0.81 to 1.09, n=50) (1.6 vs 0.9, p=0.002). To illustrate the temporal effect of maternal infection on transfer efficiency, we analysed transfer ratios of 54 symptomatic mother–infant dyads. Asymptomatic mothers were excluded from this analysis as their timing of infections cannot be concluded definitively from the timing

of PCR positivity (figure 3B). The transfer ratios based on time elapsed from the first maternal positive PCR to delivery were 0.6 (95% CI 0.39 to 1.04) (<60 days, n=22), 1.2 (95% CI 0.98 to 1.29) (60–180 days, n=27), and 0.9 (95% CI 0.33 to 2.17) (>180 days, n=5). The ratio was significantly higher in the 60–180 days group compared with the <60 days group (1.2 vs 0.6, p<0.0001). There was no significant correlation between the transfer ratio and GA at birth (Rs=0.18, p=0.1, figure 3C); however, 95% of the infants in our cohort were born at greater than 34 weeks' gestation. Transfer ratios based on the trimester of maternal infection were 0.9 (95% CI 0.39 to 1.89) (first trimester, n=7), 1.2 (95% CI 1.08 to 1.5) (second trimester, n=9), and 0.9 (95% CI 0.76 to 1.1) (third trimester, n=38) (figure 3D). The ratio was significantly higher in second trimester infections than third trimester infections (1.2 vs 0.9, p=0.02).

### Maternally derived IgG longitudinal follow-up in infants
To evaluate maternally derived IgG persistence postnatally, we followed serology in 48 infants with positive cord IgG. All infants showed a steady decrease in IgG levels over time (figure 4A). The IgG seroreversion rate was calculated for those infants who had at least one serology test during the follow-up age periods of 1–4 weeks, 5–12 weeks and 13–28 weeks. The IgG seroreversion rates for the three follow-up periods were 8% (4 of 48), 12% (3 of 25) and 38% (5 of 13), respectively. The infants who had lower levels of IgG in the cord blood became IgG negative earlier; the infants who had cord IgG levels 52–66 RFU seroreverted at 1–4 weeks, 68–150 RFU seroreverted at 5–12 weeks, and 123–251 RFU seroreverted at 13–28 weeks. Two infants who had cord IgG levels greater than 500 RFU remained seropositive at 27 weeks of age.

### Infant antibody response to perinatal SARS-CoV-2 infection
We performed surveillance serology tests at 2–4 weeks of age in 23 of 41 (56%) infants who had negative serology in the cord blood and were born to mothers with first positive PCR <14 days before delivery. Two infants showed seroconversion, including the 31-week preterm infant who tested positive for SARS-CoV-2 nasopharyngeal swab and a term infant. Interestingly, both infants were born to mothers who tested positive for SARS-CoV-2 PCR for the first time at delivery and negative for SARS-CoV-2 antibodies, indicating a new onset of infection. Both mothers were asymptomatic at delivery and remained asymptomatic for 2 weeks post-delivery. Both infants were asymptomatic for SARS-CoV-2 infection. The preterm infant, admitted to the NICU immediately after birth, was isolated from the mother for 14 days. This infant had typical respiratory symptoms for 31 weeks' prematurity and was on CPAP and nasal cannula, with 21% FiO$_2$ for 10 days. The chest X-ray did not show any evidence of infiltration. The infant did not have any symptoms or concerns attributable to COVID-19 disease during the NICU stay and was discharge home at 34 weeks and 5 days' post-menstrual age.

**Table 2** Maternal and cord blood serology and timing of maternal first positive PCR

|  | Total | 0–13 days | 14–59 days | 60–180 days | >180 days |
|---|---|---|---|---|---|
| **Maternal serology, n** | **129** | **56** | **28** | **36** | **9** |
| IgM– and IgG–, n | 45 | 35 | 4 | 5 | 1 |
| IgM+ and IgG–, n | 4 | 4 | 0 | 0 | 0 |
| IgM+ and IgG+, n | 29 | 6 | 13 | 9* | 1† |
| IgM– and IgG+, n | 51 | 11 | 11 | 22 | 7 |
| IgM+ and/or IgG+, n | 84 | 21 | 24 | 31 | 8 |
| IgM, RFU, median (range) | 27 (7–1388) | 25.5 (2–315) | 34.5 (7–1388) | 26.5 (11–263) | 25 (7–59) |
| IgG, RFU, median (range) | 84 (1–3582) | 22.5 (1–401) | 178 (1–1123) | 194.5 (22–2311) | 199 (41–3582) |
| **Cord blood serology, n** | **144** | **70** | **27** | **38** | **9** |
| IgG–, n | 61 | 48 | 8 | 4 | 1 |
| IgG+, n | 83 | 22 | 18 | 32 | 8 |
| IgG, RFU, median (range) | 66.5 (0–2916) | 14 (0–1820) | 77 (2–1164) | 232 (22–2916) | 209 (45–1173) |
| **Paired cord and maternal blood serology, n** | **125** | **54** | **26** | **36** | **9** |
| Maternal IgG+ and cord blood IgG+, n | 69 | 12 | 19 | 31 | 7 |
| Maternal IgG+ and cord blood IgG–, n | 8 | 4 | 3 | 0 | 1 |
| Maternal IgG– and cord blood IgG–, n | 45 | 37 | 4 | 4 | 0 |
| Maternal IgG– and cord blood IgG+, n | 3 | 1 | 0 | 1 | 1 |
| Maternal IgM+ and cord blood IgM+, n | 3 | 0 | 1 | 2 | 0 |
| Maternal IgM+ and cord blood IgM–, n | 29 | 10 | 11 | 7 | 1 |
| Maternal IgM– and cord blood IgM–, n | 93 | 44 | 14 | 27 | 8 |

*All nine mothers' first positive SARS-CoV-2 PCR were between 63 and 103 days before delivery.
†This mother's SARS-CoV-2 PCR was positive at 10 weeks' gestation and was positive again at the time of delivery at 39 weeks' gestation.
RFU, relative fluorescent unit.

The infant's cord blood SARS-CoV-2 PCR was negative, but nasopharyngeal PCR was positive at 24 hours of life and remained positive at discharge. Additionally, the infant's meconium and maternal blood at the time of delivery were PCR positive. The term infant roomed in with the mother in the postpartum unit and was discharged home at 2 days of life. This infants' cord blood and nasopharyngeal SARS-CoV-2 PCR were negative at 24 hours of life, and nasopharyngeal PCR was not repeated.

The term infant had the first follow-up test at 2 weeks and was found positive for IgM (225 RFU) and IgG (80 RFU) figure 4B,Cfigure 4B). The infant's IgM became negative, and IgG peaked at 1841 RFU at 8 weeks; the IgG subsequently decreased to 648 RFU at 24 weeks. The preterm infant showed serial negative serology tests after birth on days 2, 4, and 8, then seroconverted on day 16 (IgM 1548 RFU, IgG 335 RFU) (figure 4C). The infant's IgM decreased to 134 RFU, and IgG increased to 1873 RFU at 8 weeks.

## DISCUSSION

We conducted a prospective observational study in 145 pregnant mothers with SARS-CoV-2 infections during pregnancy and 147 of their infants. The majority of infected mothers seroconverted before delivery. The IgG

levels in maternal blood at delivery and cord blood were highly correlated. High transplacental IgG transfer ratios were observed when infection onset was greater than 60 days prior to delivery or in the second trimester. The persistence of maternal-derived IgG in infants was positively correlated to the initial cord blood level. Additionally, we showed strong antibody responses to perinatal SARS-CoV-2 infection in two asymptomatic neonates.

This study was conducted from April 2020 to March 2021 when Northern California experienced the peaks of COVID-19 pandemic outbreaks. During this period, our universal screening programme in labour and delivery units identified 6.5% of mothers who had at least one positive SARS-CoV-2 PCR during their current pregnancy. It is possible that positive cases would have been missed as some asymptomatic or mildly symptomatic mothers were not tested. The majority of mothers had asymptomatic or mild-moderate infections, consistent with previous cohort studies.[12 16] The maternal IgG levels at delivery were relatively low, comparable with levels in non-intensive care unit patients.[32] Importantly, the temporal profiles of maternal and cord blood IgG levels were in parallel, peaking around 60–120 days post-maternal infection. The timing and efficiency of maternal IgG transfer have important implications for developing

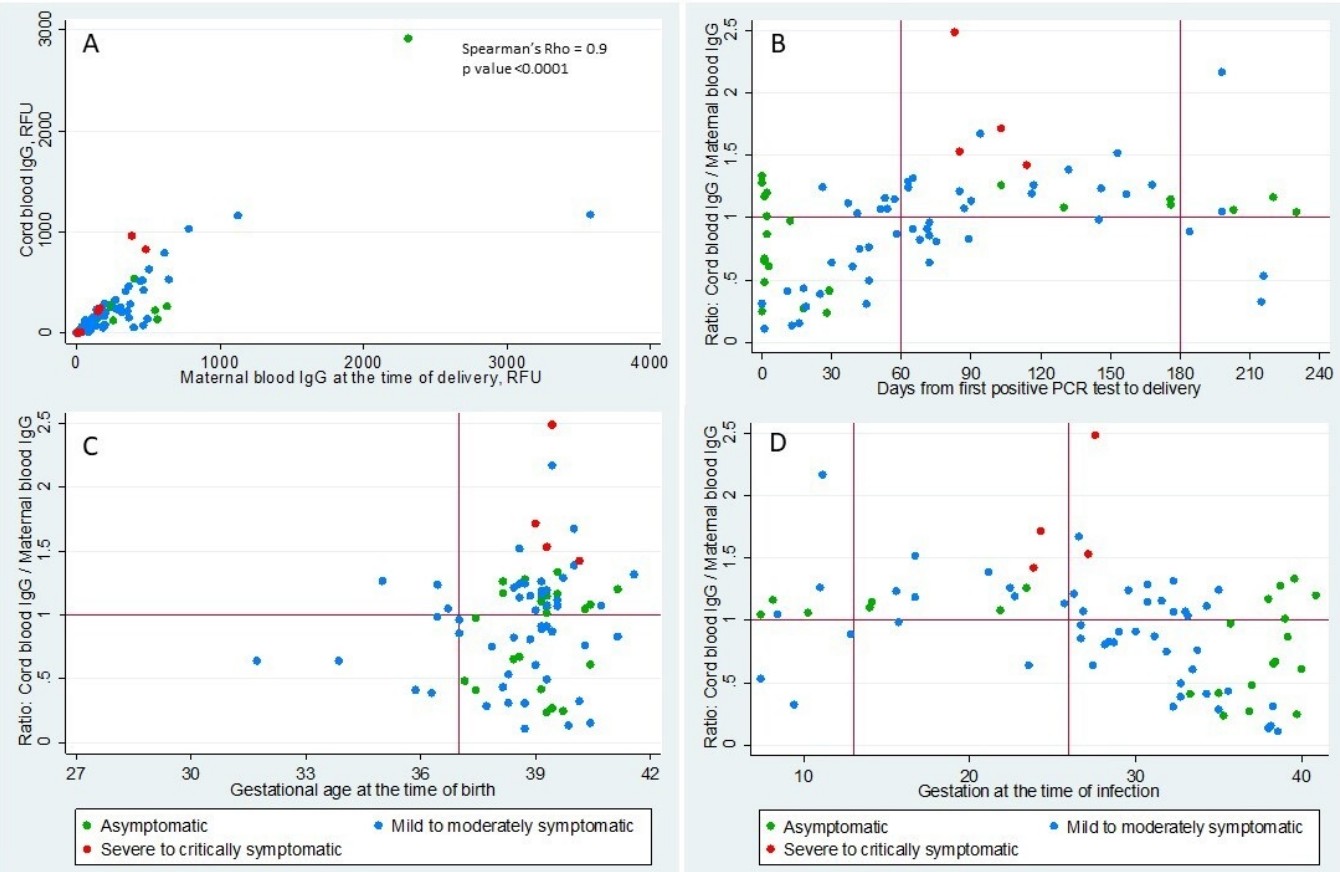

**Figure 3** Correlation of cord blood and maternal IgG and distribution of IgG transplacental transfer ratio. (A) Scatterplot shows the correlation of cord blood SARS-CoV-2 IgG levels in y-axis and maternal blood SARS-CoV-2 IgG levels in x-axis in relative fluorescent unit (RFU). (B) Scatterplot shows the distribution of IgG transplacental ratio (cord blood/maternal blood SARS-CoV-2 IgG levels) in the y-axis and days from maternal first positive SARS-CoV-2 PCR test to delivery in x-axis. (C) Scatterplot shows the distribution of IgG transplacental ratio in the y-axis and gestational age at the time of delivery in x-axis. (D) Scatterplot shows the distribution of IgG transplacental ratio in the y-axis and gestational age at the time of maternal first positive SARS-CoV-2 PCR test in x-axis. The different colours represent the severity of the maternal symptoms at the time of diagnosis.

maternal immunisation strategies to protect infants.[33–35] For example, in maternal pertussis immunisation, infant seropositivity rate and cord blood IgG levels to pertussis toxin were higher following Tdap immunisation during the second trimester than during the third trimester. We studied pregnant mothers who had SARS-CoV-2 infections in all three trimesters and provide a comprehensive profile of transplacental IgG transfer with respect to the timing of infections throughout gestation. We observed that transfer ratio was 0.6 when infection onset was less than 60 days before delivery; plateaus at 1.2 and 0.9 when infections occurred 60–180 days and greater than 180 days before delivery. Prior studies of pregnancy-related infection in the last 70 days of gestation found impaired SARS-CoV-2 IgG transplacental transfer (ratio 0.7).[7 8] Another study characterised a cohort of pregnant mothers who had infections during the last 120 days of gestation and showed that transfer ratios increased with length of time from infection to delivery, with transfer ratios reached above 1.0 in the majority of mothers.[6] Taken together, these studies demonstrate that cross-placental SARS-CoV-2 IgG transfer occurs throughout

gestation, and a higher transfer efficiency is achieved when infection onset is more than 2 months prior to delivery. Matching the peak IgG transplacental transfer and the peak immune response after maternal infection may result in high cord IgG. Information from these maternal and cord serology studies is helpful to inform the timing of maternal vaccination in pregnancy to optimise neonatal immunity in concert.

While the persistence of maternal-derived IgG in infants showed a wide range, from 2 weeks to more than 26 weeks of age, the patterns of IgG decay in these infants were very similar. An important observation is that IgG positivity in infants is positively associated with the initial cord IgG levels that are determined by maternal IgG levels and transplacental transfer efficiency. As more pregnant mothers are vaccinated for SARS-CoV-2, knowledge of passive immunity in infants may inform mother–infant care and SARS-CoV-2 protective strategy in infants.

Consistent with prior literature showing rare vertical maternal–fetal transmission,[18–21] we found only one infant with confirmed intrapartum-acquired neonatal infection.[21] This infant was seronegative in cord blood

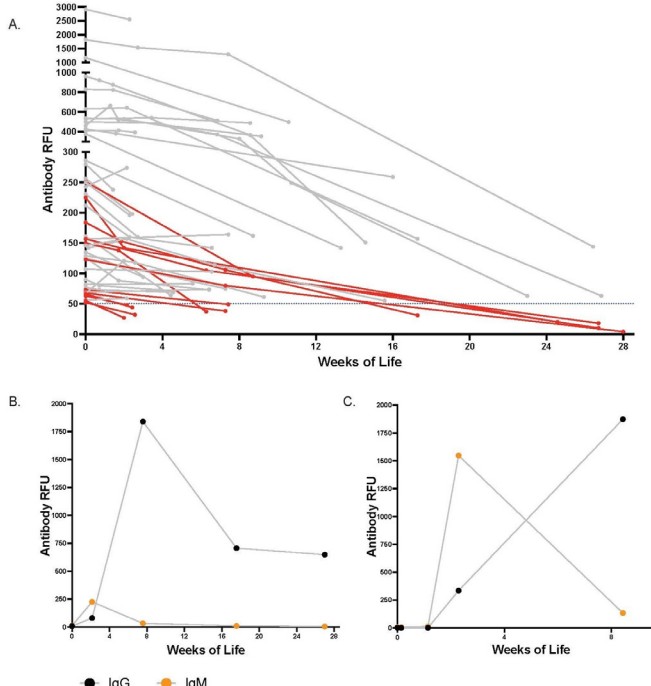

**Figure 4** Longitudinal follow-up of SARS-CoV-2 antibody levels in infants. (A) The longitudinal IgG levels of the infants who had cord blood IgG level >50 relative fluorescent unit (RFU). The infants' IgG levels in RFU is shown in y-axis, and the age of the infant in weeks at the time of follow-up is shown in x-axis. The infants whose IgG became negative, <50 RFU, during the longitudinal follow-up are shown in red colour. (B) The IgG and IgM levels of the term infant whose cord antibody was negative and seroconverted at 2 weeks of life. (C) The IgG and IgM levels of the 31 weeks' preterm infant with confirmed intrapartum SARS-CoV-2 infection whose cord antibody was negative and seroconverted at 2 weeks of life.

and during the first week of life but seroconverted at 2 weeks of life, providing insight into the timing of infant seroconversion in the setting of intrapartum infection. We identified another infant who seroconverted at 2-week follow-up test; however, available virology and serology data are not sufficient to determine the timing and mode of this perinatal infection. Clinical presentations of perinatal SARS-CoV-2 infection have been described previously;[17 22 36] however, little is known about neonatal serology response and long-term clinical outcomes. Interestingly, both infants in our study had asymptomatic infection but mounted strong antibody responses; the timing of seroconversion and levels of IgM and IgG are comparable with that observed in adult patients with severe disease.[32] Both infants remained asymptomatic in the first months of life. Their long-term clinical outcomes, along with immune status, will be followed. Additionally, these two cases highlight the increased risk of perinatal SARS-CoV-2 infection in infants born to mothers who have new-onset infections around the time of delivery,[17] with implications for developing targeted protection measures and post-natal antibody screening for high-risk newborns.

In our study, three infants were positive for IgM in cord blood but negative for SARS-CoV-2 virologically in birth specimens and negative for IgM and IgG at 2 and 3 weeks of age, suggesting these transient IgM levels may be false positives or maternal blood contamination. There were two prior case reports describing similar transient positive IgM levels in the cord blood without virological evidence of infection.[37 38] Thus, diagnosis of congenital SARS-CoV-2 infection cannot be made based solely on the presence of IgM in the cord blood.[39–42]

This maternal–infant serology study, one of the largest cohorts to date, included pregnant mothers with SARS-CoV-2 infection in all three trimesters of pregnancy and provided a more comprehensive understanding of maternal SARS-CoV-2 IgG transplacental transfer. This is the first longitudinal study that has followed the level of maternally derived SARS-CoV-2 IgG in infants up to 28 weeks and neonatal serology response after perinatal SARS-CoV-2 infection up to 24 weeks. Another strength of the study is that the cohort is representative of COVID-19 in the community. Over 90% of the mothers in this cohort are Hispanic, a population highly impacted by the COVID-19 pandemic.

Our study has several limitations. It was conducted in a single healthcare system. Our cohort had few severe cases and premature births before 35 weeks of gestation. Our longitudinal infant serology follow-up had significant attrition and the timing of blood sampling was variable due to the challenges of coming to the clinics during the pandemic. The timing of maternal infection was based on the first positive PCR. In asymptomatic mothers with first positive SARS-CoV-2 PCR at the time of delivery, we were unable to ascertain the precise timing of infection. Universal screening at the time of admission also introduces a bias in the identification of asymptomatic SARS-CoV-2 cases at or near-term gestation, as the universal screening was not implemented in our prenatal care visits and asymptomatic screening was not readily available in our general community during the study period.

## CONCLUSION

Our study provides insights into the intricate connections between the timing of maternal SARS-CoV-2 infection, dynamics of maternal antibody production and transplacental immunity transfer. These processes determine the level of maternally derived IgG in infants at birth, which in turn affects persistence of passive immunity in infants. Neonates are capable of mounting strong serology responses to perinatal SARS-CoV-2 infection. These findings have important implications in determining optimal timing of vaccination in pregnant mothers. Future investigations are needed to address the durability and protection of passively and actively acquired antibodies in the infant.

**Author affiliations**
[1]Department of Pediatrics, Division of Neonatology, Santa Clara Valley Medical Center, San Jose, California, USA

[2]Department of Pediatrics, Stanford University School of Medicine, Stanford, California, USA

[3]Department of Pediatrics, Division of Pediatric Infectious Diseases and Global Health, University of California San Francisco, San Francisco, California, USA

[4]Division of Maternal-Fetal Medicine, Department of Obstetrics, Gynecology and Reproductive Sciences, University of California San Francisco, San Francisco, California, USA

[5]Department of Family Medicine, Stanford University School of Medicine, Stanford, California, USA

[6]Department of Laboratory Medicine, University of California San Francisco, San Francisco, California, USA

[7]Department of Medicine, University of California San Francisco, San Francisco, California, USA

[8]Department of Pathology, Santa Clara Valley Medical Center, San Jose, California, USA

[9]Department of Pediatrics, Division of Pediatric Hospital Medicine, Santa Clara Valley Medical Center, San Jose, California, USA

[10]Department of Pediatrics, Marshall University, Huntington, West Virginia, USA

[11]Department of Obstetrics and Gynecology, Division of Maternal-Fetal Medicine, Santa Clara Valley Medical Center, San Jose, California, USA

[12]Department of Obstetrics and Gynecology, Stanford University School of Medicine, Stanford, California, USA

**Acknowledgements** Thank you to Dr Margaret Feeney for support of these experiments and Robin D Wu who reviewed the manuscript and provided helpful comments and suggestions. We thank the mothers, newborns and their families who participated in the study; staff and providers in labour and delivery, postpartum unit, NICU, pathology and laboratory services, at Santa Clara Valley Medical Center and O'Connor Hospital; outpatient paediatrics clinic, and BRIDGE home follow-up programme. We thank the First 5 of Santa Clara County for their support.

**Contributors** DS conceptualised and designed the project; participated in patient enrolment, sample collection, data visualisation and interpretation; and wrote the manuscript draft. PJ conceptualised and designed the project; participated in patient enrolment, sample collection, data collection, analysis, and interpretation; and edited the manuscript. MP and SLG designed collection and processing protocols, performed sample processing, oversaw experiments and data analysis, provided funding and edited the manuscript. SRN conceptualised and designed the project; participated in patient enrolment, sample collection, and data collection; and edited the manuscript. DSR participated in study design, patient enrolment and sample collection. AH coordinated data collection and management, participated in sample collection and oversaw the implementation of the project. CVF participated in patient enrolment, and coordinated sample collections, processing, and data management. LW designed collection and processing protocols, and performed sample processing and data collection. JL and CBTN, CYL, UJ and VJG designed and performed experiments, and performed sample processing and data collection. PC, LF, GRA and AV optimised collection and processing protocols, and performed sample processing and data collection. AW designed and oversaw serology assays. EA, PN and CM oversaw sample collection and processing. CA, SM, MS, MC, JM, SA and NM participated in patient enrolment and sample collection. MN participated in data analysis and preparing the figures for the manuscript. RP participated in the study implementation and sample collection. JB designed and oversaw the patient recruitment and implementation of the project. All authors reviewed and approved the manuscript.

**Funding** This work was supported by Bill and Melinda Gates Foundation (INV-017035 to SLG), National Institutes of Health (NIAID K08AI141728 to SLG, NIAID K23AI127886 to MP), Marino Family Foundation, and Valley Medical Center Foundation.

**Disclaimer** The funders did not have any role in the study design, patient recruitment, data collection, analysis or interpretation of the results. All the authors had full access to the full data in the study and accept responsibility to submit for publication.

**Competing interests** None declared.

**Patient consent for publication** Not required.

**Ethics approval** The study obtained ethics approval from the Institutional Review Board of Santa Clara Valley Medical Center (IRB reference #20-021). Patients provided written informed consent prior to study enrolment and all study procedures.

**Provenance and peer review** Not commissioned; externally peer reviewed.

**Data availability statement** Data are available in a public, open access repository. De-identified data are published in Mendeley data sharing site and available in the following: doi:10.17632/6scfwt55fd.2.

**ORCID iDs**
Stephanie L Gaw http://orcid.org/0000-0003-0891-6964
Priya Jegatheesan http://orcid.org/0000-0003-4068-0330

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
