## [Reviewer comments · BMJ Open]

ARTICLE DETAILS

TITLE (PROVISIONAL)	Passive and active immunity in infants born to mothers with SARS-CoV-2 infection during pregnancy: Prospective cohort study
AUTHORS	Song, Dongli; Prah, Mary; Gaw, Stephanie; Narasimhan, Sudha Rani; Rai, Daljeet; Huang, Angela; Flores, Claudia; Lin, Christine; Jigmeddagva, Unurzul; Wu, Alan; Warriar, Lakshmi; Levan, Justine; Nguyen, Catherine; Callaway, Perri; Farrington, Lila; Acevedo, Gonzalo; Gonzalez, Veronica; Vaaben, Anna; Nguyen, Phuong; Atmosfera, Elda; Marleau, Constance; Anderson, Christina; Misra, Sonya; Stemmler, Monica; Cortes, Maria; McAuley, Jennifer; Metz, Nicole; Patel, Rupalee; Nudelman, Matthew; Abraham, Susan; Byrne, James; Jegatheesan, Priya

VERSION 1 – REVIEW

REVIEWER	Gao, Ya-dong Wuhan University
REVIEW RETURNED	16-May-2021

GENERAL COMMENTS	This is a very important study to demonstrate the IgG positivity in infants born to mothers infected with SARS-CoV-2 during pregnancy. The results may guide the vaccination strategy for pregnant women in risk. I have only one comment: the results are presented mainly in scattered plots; this make the difference of IgG levels among different trimesters not clear at a glance. When possible, descriptive histogram will help to illustrate the differences.
--

REVIEWER	O'Ryan, Miguel University of Chile
REVIEW RETURNED	19-May-2021

GENERAL COMMENTS	The study from Dongli and colleagues describes three aspects related to SARS-COV-2 infection in pregnant women and their offsprings: IgG and IgM maternal seroresponses after a positive PCR during pregnancy, magnitude of transplacental passage measured by the ratio of cord blood/maternal antibodies with an analysis of this transfer in relation to the time of infection, and the duration of antibodies in the newborn, including two cases of perinatal infection. For this a cohort of 145/147 mother-newborn pairs are prospectively included with the characteristic of having had a positive PCR sometime during pregnancy, with and without symptoms. A blood sample was obtained from the mother and cord at the time of delivery and then at different time points in the newborn infants. The study provides important insights of which the most novel, but not unexpected, is the relationship between the interval between infection and birth with cord blood positivity. Other
---

findings including seroreversion over time in the infant directly proportional to the antibody levels at birth and the seroresponse curves in the two likely perinatal infections are interesting and novel, but less robust as numbers are small.

Major comments:

- 1) The main question that rises while reading the manuscript is if this was a structured protocol with aims and hypothesis and sample size calculation or not.
- 2) The description of the methodology is quite general which raises some important questions
 - a) The timing of PCR testing during pregnancy is confusing as it is mentioned that “universal screening protocol for SARS-CoV-2 infection in women presenting in labor or within three days prior to admission for elective deliveries” was implemented. If this was the case, I am not clear as to how women with positive PCRs were recruited well in advance of delivery (unless I misunderstood).
 - b) Was there a protocol for PCR testing in symptomatic women and another for asymptomatic women? (screening). The manuscripts states “SARSCoV-2 infection was diagnosed based on a positive SARS-CoV-2 reverse transcriptase polymerase chain reaction (RT-PCR) test using nasopharyngeal specimen performed either before delivery or through universal screening at delivery”; but it is unclear how and when this was performed.
 - c) The collection of infant blood samples does not seem to have been protocolized as different number of samples and timepoints for sampling are shown.
 - d) Was there any expectation as to how many potentially infected infants were to be recruited?

Other comments

Abstract: Page 5

- 1) Line 18: should state that women could be symptomatic or asymptomatic
- 2) Line 22: the description of the two cases is confusing as placed here, would suggest to include at the end.
- 3) Line 41: seroreversion may be a better term than “negative conversion”.
- 4) Line 43: The reducing number of tested infants as weeks go by needs to be addressed.
- 5) Line 42: Suggest to avoid the word “protect” as this is unknown; better maybe “persists”
- 6) Should be “Two neonates mounted.....”

Strengths and limitations: Page 6

- 7) Line 18: stress that it is in 2 cases
- 8) Line 21: This statement does not seem to be supported by this study

Introduction

9) First paragraph: The relevance of Covid-19 during pregnancy in magnitude and severity should be briefly discussed. In addition the impact in newborns should also be presented. Current knowledge indicates increased severity in mothers but this is less clear among infants. The rationale of vaccinating mothers to protect the infant, thus far, can be questioned. As this is an important rationale for this study as vaccination during pregnancy is discussed, including the

potential optimal timing, due to the findings of this study, this should be introduced. The same for persistence of antibodies in the newborn, one could question why it is important? (Line 39).

Methods: (main issues mentioned above)

10) Protocol for PCR testing, protocol for sample collection and processing? For mothers and newborns.

11) In which subset of infants was PCR performed and for which aim?

12) Data collection and analysis: any sample size calculation?

Results

13) Page 9, line 43-45: As mentioned, the testing period during pregnancy is unclear. Could positive cases have been missed?

14) Page 10, line 18: Where there more than one positive PCR test performed? If so, did this influence seropositivity?

15) Line 37: The 3 IgM positive infants were born to IgM positive mothers. Were there any IgM positive mothers that did not have IgM positive infants?. (IgMs are not shown in tables).

16) Was it confirmed that these infants were fully asymptomatic?

17) Page 11, line 10: Was transfer of antibodies in symptomatic mothers related to antibody levels?

18) Line 26: It seems that authors are describing the same phenomenon in a different manner (days vs trimesters).

19) Line 41: Seroreversion seems better than seroconversion here and in general as this is a downtrend

20) Line 45: Why were the numbers significantly reduced? The protocol for obtaining blood sample(s) needs to be clearly described.

21) Line 50: which three follow-up periods? This is confusing as it seems that titers of children who seroreverted were higher in older infants?

22) Page 12, line 12: this was only nasopharyngeal testing?

23) Line 17: Did these mothers develop symptoms later? Did they have a history of contact exposure?

24) Line 23: any symptom at all?

25) Line 30: Was this mother definitely confirmed as an asymptomatic infection?

26) line 37-48: clinical status as detailed as possible in relation to any possible Covid19 related involvement would be good.

Discussion

27) Page 13, line 28-29: Do infants actually have to be protected, through vaccination, with such a low risk? This is repeated several times throughout the discussion and probably should be argued.

28) Line 37: CI for these risks would be important due to the relatively low numbers.

29) Page 14: Line 6-8: Once again, it is unclear if vaccination will be required to protect newborns.

30) Table 1: Meaning of "Gravida" and "Para"; Race (do not include n(%)); Symptomatic at the time of diagnosis is redundant

31) Table 2: I suggest Weeks or Days but not both; Data on IgM?

32) Figure 2: Panel D seems like the inverse of Panel B? is it needed?

33) Figure 3: Sampling points seem quite disperse? Need to explain

VERSION 1 – AUTHOR RESPONSE

Reviewer: 1

Dr. Ya-dong Gao, Wuhan University

Comments to the Author:

This is a very important study to demonstrate the IgG positivity in infants born to mothers infected with SARS-CoV-2 during pregnancy. The results may guide the vaccination strategy for pregnant women in risk.

I have only one comment: the results are presented mainly in scattered plots; this make the difference of IgG levels among different trimesters not clear at a glance. When possible, descriptive histogram will help to illustrate the differences.

We appreciate your suggestion. To illustrate the difference in maternal IgG levels at delivery among mothers with different timing of infection, we created the boxplots below comparing maternal IgG levels between different intervals (1-13d, 14-59d, 60-179d, > 180d) from maternal infection to delivery. In order to be consistent with the description of the IgG levels presented in Table 2, we chose the timing of maternal infection groups instead of trimester of infection.

Reviewer: 2

Dr. Miguel O’Ryan, University of Chile

Comments to the Author:

The study from Dongli and colleagues describes three aspects related to SARS-COV-2 infection in pregnant women and their offsprings: IgG and IgM maternal seroresponses after a positive PCR during pregnancy, magnitude of transplacental passage measured by the ratio of cord blood/maternal antibodies with an analysis of this transfer in relation to the time of infection, and the duration of antibodies in the newborn, including two cases of perinatal infection. For this a cohort of 145/147 mother-newborn pairs are prospectively included with the characteristic of having had a positive PCR sometime during pregnancy, with and without symptoms. A blood sample was obtained from the mother and cord at the time of delivery and then at different time points in the newborn infants. The study provides important insights of which the most novel, but not unexpected, is the relationship between the interval between infection and birth with cord blood positivity. Other findings including seroreversion over time in the infant directly proportional to the antibody levels at birth and the seroresponse curves in the two likely perinatal infections are interesting and novel, but less robust as numbers are small.

Major comments:

1) The main question that rises while reading the manuscript is if this was a structured protocol with aims and hypothesis and sample size calculation or not.

We designed this prospective observational study in March 2020, when little was known about the impact of SARS-CoV-2 infection on pregnant mothers and their infants. Literature from other infectious diseases have shown that timing of maternal infection/vaccination affects the efficiency of maternal IgG transplacental IgG transfer. Our study was designed to investigate (1) maternal SARS-CoV-2 antibody transplacental transfer with respect to the timing of maternal infection during gestation, (2) antibody response to SARS-CoV-2 infection in the newborns, and (3) persistence of passively- and actively-acquired SARS-CoV-2 antibodies in infants.

We did not perform sample size calculation when we designed the study protocol because there were no reference range values available for maternal, cord blood, or neonatal antibodies at the beginning

of the pandemic.

2) The description of the methodology is quite general which raises some important questions

a) The timing of PCR testing during pregnancy is confusing as it is mentioned that “universal screening protocol for SARS-CoV-2 infection in women presenting in labor or within three days prior to admission for elective deliveries” was implemented. If this was the case, I am not clear as to how women with positive PCRs were recruited well in advance of delivery (unless I misunderstood).

b) Was there a protocol for PCR testing in symptomatic women and another for asymptomatic women? (screening). The manuscripts states “SARSCoV-2 infection was diagnosed based on a positive SARS-CoV-2 reverse transcriptase polymerase chain reaction (RT-PCR) test using nasopharyngeal specimen performed either before delivery or through universal screening at delivery”; but it is unclear how and when this was performed.

Thank you for these important questions regarding our study eligibility criteria and patient recruitment process. We have revised our manuscript in the Methods section to provide better descriptions of screening and PCR testing protocols in our Labor and delivery units:

“On April 15th, 2020, our institution implemented universal SARS-CoV-2 screening protocol in Labor and Delivery units. All women admitted for delivery or within three days before admission for elective deliveries were tested for SARS CoV-2 by PCR using a nasopharyngeal swab. From October 2020 onwards, women who tested positive within 90 days prior to admission for delivery and did not have new symptoms of COVID were not retested at the time of delivery per our local county public health department recommendations. In addition to PCR testing, mothers were screened for a history of SARS CoV-2 infection and PCR testing during pregnancy. PCR test was done anytime during pregnancy if the mother experienced symptoms concerning COVID-19 or had close contact with a person with COVID-19.

c) The collection of infant blood samples does not seem to have been protocolized as different number of samples and timepoints for sampling are shown.

Serial infant blood samples were initially designed to be collected at two weeks, two months, and six months coordinated with routine pediatric clinic visits. During the pandemic the visit schedules varied significantly due to parental hesitance to come to the clinics for concerns of COVID exposure. Thus, infants’ blood samples were collected anytime between 1-4 weeks, 5-12 weeks, and 13-28 weeks at the time of clinic visits.

We have revised the description of our infant blood sample collection protocol in the Methods section and added this information as a limitation in the Discussion section.

d) Was there any expectation as to how many potentially infected infants were to be recruited?

We did not have an expected number of infected infants as the incidences of SARS-CoV-2 infection in pregnant mothers and newborns were unknown at the beginning of the pandemic, when the protocol was developed.

Other comments

Abstract: Page 5

1).Line 18: should state that women could be symptomatic or asymptomatic.

We made the suggested change in the abstract.

2) Line 22: the description of the two cases is confusing as placed here, would suggest to include at the end.

We moved this sentence to the end of the results section as suggested.

3) Line 41: seroreversion may be a better term than “negative conversion”.

We changed “negative conversion” to “seroreversion” as suggested.

4) Line 43: The reducing number of tested infants as weeks go by needs to be addressed.

Due to the word limits in the abstract we addressed this in the main text results and discussion.

5) Line 42: Suggest to avoid the word “protect” as this is unknown; better maybe “persists”

We changed “protect” to “persist” in the abstract conclusion as suggested.

6) Should be “Two neonates mounted.....”.

We changed “Neonates mount” to “two neonates mounted” as suggested.

Strengths and limitations: Page 6

7) Line 18: stress that it is in 2 cases.

We specified that the seroconversion was in 2 neonates as suggested.

8) Line 21: This statement does not seem to be supported by this study.

In our study, the timing of SARS CoV-2 infection was based on the time of first maternal positive PCR. There were 31 asymptomatic mothers whose infections were identified by positivity in screening PCR at the time of delivery. Ten of these mothers tested positive for SARS CoV-2 IgG but negative for IgM at the time of delivery. Therefore, in these asymptomatic mothers the timing of the first positive PCR might not represent the precise timing of infection. We have revised the statement as “In asymptomatic mothers with first positive SARS-CoV-2 PCR at the time of delivery, we were unable to ascertain the precise timing of infection.”

We have added this detail to the Results section and discussed it as a study limitation.

Introduction

9) First paragraph:

The relevance of Covid-19 during pregnancy in magnitude and severity should be briefly discussed. In addition the impact in newborns should also be presented. Current knowledge indicates increased severity in mothers but this is less clear among infants.

The rationale of vaccinating mothers to protect the infant, thus far, can be questioned. As this is an important rationale for this study as vaccination during pregnancy is discussed, including the potential optimal timing, due to the findings of this study, this should be introduced.

The same for persistence of antibodies in the newborn, one could question why it is important? (Line 39).

We have revised the Introduction section to include more information on maternal and neonatal/infant SARS-CoV-2 infection to support the rationale for our study.

Methods: (main issues mentioned above)

10) Protocol for PCR testing, protocol for sample collection and processing? For mothers and newborns.

Maternal and neonatal nasopharyngeal samples were collected according to our hospital standard procedures. PCR tests were performed by hospital clinical laboratories using the following four assays that have been validated and used for the clinical diagnostic purposes: Xpert® Xpress SARS-CoV-2 assay (Cepheid, Sunnyvale, California, USA), DiaSorin Simplexa™ COVID-19 Direct assay (Diasorin Molecular, Cypress, California, USA), Perkin Elmer® nCoV NAD assay (Perkin Elmer, Waltham, Massachusetts, USA), and Hologic® Aptima™ SARS-CoV-2 Assay (Hologic Inc., Marlborough, Massachusetts, USA). We have added this information in the Methods section and supplemental Methods section.

11) In which subset of infants was PCR performed and for which aim?

Newborns born to mothers who were positive within 14 days of delivery (within their contagious period) were tested using a nasopharyngeal swab for SARS-CoV-2 PCR to identify newborns with perinatal SARS-CoV-2 infection and describe their antibody response (specific aim 3).

12) Data collection and analysis: any sample size calculation?

We did not perform sample size calculation. See response to question #1.

Results

13) Page 9, line 43-45: As mentioned, the testing period during pregnancy is unclear. Could positive cases have been missed?

We have revised the methods section to clarify the testing period during pregnancy (see response to question #2). It is possible that positive cases would have been missed if infected mothers were not SARS-CoV-2 PCT tested during pregnancy, especially the asymptomatic or mildly symptomatic infections. We have added this in the Discussion Section.

14) Page 10, line 18: Where there more than one positive PCR test performed? If so, did this influence seropositivity?

There were 16 mothers who had more than one positive PCR test and 15 of those had maternal serology available. Of the 15, 12 were positive for IgG and/or IgM at the time of delivery. The seropositivity in these 15 women was 80% compared to the 65% in our overall study cohort.

15) Line 37: The 3 IgM positive infants were born to IgM positive mothers. Were there any IgM positive mothers that did not have IgM positive infants? (IgMs are not shown in tables).

Of the 33 mothers who were IgM positive only 3 infants were IgM positive, the other 30 infants were IgM negative in the cord blood. The Cord IgM results are added in the revised Table 3.

16) Was it confirmed that these infants were fully asymptomatic?

Two of these were term infants and had an asymptomatic newborn course in the hospital and

remained asymptomatic confirmed during outpatient pediatric visit in the first month of life. The third infant was a 31 week gestational age premature infant who was delivered due to in utero growth restriction. This infant had typical respiratory symptoms due to lung immaturity and was on CPAP and nasal canula, with 21% FiO₂ for 3 weeks. The chest X ray did not show any evidence of infiltration. We added this clinical information in the Results section.

17) Page 11, line 10: Was transfer of antibodies in symptomatic mothers related to antibody levels?

In symptomatic mothers, there was no correlation between the maternal IgG level and the transfer of antibodies (i.e. ratio of cord blood IgG to maternal IgG) (R_s 0.07, $p=0.6$).

18) Line 26: It seems that authors are describing the same phenomenon in a different manner (days vs trimesters).

We described antibody transfer ratios based on the intervals (days) of the maternal infection prior to delivery (Figure 3 B), the method used in two previous published studies. In addition, we described the antibody transfer ratios based on the gestational age at the time of infection (Figure 3D). Mothers who have the same number of days between infection and delivery may be infected at different gestational ages if they are delivered at different gestational ages. Thus, we included Figure 3D to show that infection at different gestational ages had different transfer ratios. We performed statistical analysis to compare the transfer ratios for the maternal infections in 3 different trimesters. We think that this information adds additional value for clinicians who commonly evaluate infection in the context of gestational age / trimester.

19) Line 41: Seroreversion seems better than seroconversion here and in general as this is a downtrend

We have changed the negative “serovonversion” to “seroreversion” in the manuscript.

20) Line 45: Why were the numbers significantly reduced?

During the pandemic the parents were reluctant to come to the clinics and some parents declined to continue to have blood draw from their babies for study purpose.

The protocol for obtaining blood sample(s) needs to be clearly described.

Serial infant blood samples were intended to be collected at two weeks, two months, and six months coordinated with routine pediatric visits. However, during the pandemic routine pediatric visit schedules varied significantly due to parental reluctance to come to the clinics. Hence, the blood samples were collected anytime between 1-4 weeks, 5-12 weeks, and 13-28 weeks at the time when infants came for clinic visits.

We described the intended study protocol in the Methods section and explained the reason for variable blood sampling time in the Discussion section.

21) Line 50: which three follow-up periods? This is confusing as it seems that titers of children who seroreverted were higher in older infants?

We have clarified this sentence as “The infants who had lower levels of IgG in the cord blood became IgG negative earlier; the infants who had cord IgG levels were 52-66 RFU seroreverted at 1-4 weeks, 68-150 RFU seroreverted at 5-12 weeks, and 123-251 RFU seroreverted at 13-28 weeks.”

Our observation in Figure 4 shows that the infants who have a higher cord blood IgG level take a longer time to become negative. Hence the age of the infant at the time of seroreversion is older for those with higher initial cord blood titer.

22) Page 12, line 12: this was only nasopharyngeal testing?

Yes, this was by nasopharyngeal testing. We have clarified this in the manuscript.

23) Line 17: Did these mothers develop symptoms later? Did they have a history of contact exposure?

Both the mothers were asymptomatic at delivery and remained asymptomatic for 2 weeks post delivery. , i.e. they were not in the pre-symptomatic phase at the time of delivery.

The mother of the term infant had a history of contact exposure before delivery and the mother of the preterm infant did not have any history of contact exposure.

24) Line 23: any symptom at all?

The 31 weeks preterm infant with the perinatal SARS-COV-2 infection did not have any symptoms that was concerning for COVID infection. This infant had typical respiratory symptoms for 31 weeks prematurity and was on CPAP and nasal canula, with 21% FiO₂ for 10 days. The chest X ray did not show any evidence of infiltration. The infant did not have any symptoms or concerns attributable to COVID-19 disease during the NICU stay and was discharge home at 34 weeks and 5 days post menstrual age. We have added the above details in the results section.

25) Line 30: Was this mother definitely confirmed as an asymptomatic infection?

This mother's nasopharyngeal PCR was positive at the time of delivery and she had the history of close contact exposure to people with COVID. She was confirmed to be asymptomatic for 2 weeks after delivery.

26) line 37-48: clinical status as detailed as possible in relation to any possible Covid19 related involvement would be good.

The 31 weeks preterm infant with the perinatal SARS-COV-2 infection did not have any symptoms that was concerning for COVID infection. This infant had typical respiratory symptoms for 31 weeks prematurity and was on CPAP and nasal canula, with 21% FiO₂ for 10 days. The chest X ray did not show any evidence of infiltration. The infant did not have any symptoms or concerns attributable to COVID-19 disease during the NICU stay and was discharge home at 34 weeks and 5 days post menstrual age. We have added the above details in the results section.

Discussion

27) Page 13, line 28-29: Do infants actually have to be protected, through vaccination, with such a low risk? This is repeated several times throughout the discussion and probably should be argued.

This statement "The timing and efficiency of maternal IgG transfer have important implications for developing maternal immunization strategies to protect infants.17-19" is referring to what is shown in literature in other infectious diseases and serves as a rationale for studying the association between timing of maternal SARS-CoV-2 infection and antibody transfer.

Several studies have shown that infant born to mothers with SARS-CoV-2 infection have a low risk for

SARS-CoV-2 infection and about 50% of infected neonates are symptomatic. Emerging evidence from a large-scale international investigation (Villar, 2021) showed that infants born to women with COVID-19 during pregnancy had significantly higher risk for the severe perinatal morbidity and mortality. These risks remained significant after adjusting for prematurity, indicating a direct effect of COVID-19 on the infants. Furthermore, infants (age <1 year) might be at increased risk for severe illness from SARS-CoV-2 infection. This suggests that there is a role for maternal vaccination to prevent perinatal mortality and morbidity in newborns and infants. We have added the background information in the introduction section to support the rationale for our study.

28) Line 37: CI for these risks would be important due to the relatively low numbers.

We have added the 95% CI for ratios based on maternal symptoms group, infection time and trimester of infection in the results section.

29) Page 14: Line 6-8: Once again, it is unclear if vaccination will be required to protect newborns.

Please see response to question #27.

We have revised this sentence in the manuscript to state "Information from these maternal and cord serology studies might be helpful to guide the timing of maternal vaccination in pregnancy to optimize neonatal immunity in concert."

30) Table 1: Meaning of "Gravida" and "Para"; Race (do not include n(%)); Symptomatic at the time of diagnosis is redundant

We have defined the mean of gravida as "number of pregnancies" and para as "number of deliveries" in the footnote of the table.

We removed the n(%) next to race and symptomatic at the time of diagnosis from Table 1.

31) Table 2: I suggest Weeks or Days but not both; Data on IgM?

We have removed the weeks from column heading of Table 2. Data on IgM has been added to table 2 for the paired maternal and cord blood samples.

32) Figure 2: Panel D seems like the inverse of Panel B? is it needed?

We show the transfer ratios based on the duration (days) of the maternal infection prior to delivery in panel B. In addition, we show the ratios based on the gestational age at the time of infection in panel D.

If the deliveries happened at the same gestation for all infants, the panel D would be inverse of panel B. However, in our cohort there is variation in gestational age at delivery and thus panel D provides additional information regarding the effect of the gestational age at the time of maternal infection on antibody transfer ratio.

Please see examination for question #18.

33) Figure 3: Sampling points seem quite disperse? Need to explain

Figure 3 A shows correlation of cord blood and maternal blood IgG at the time of delivery. Figure 3B, C and D present antibody transfer ratios that did not involving sampling. We think this comment

referred to Figure 4, panel A – sampling points of the longitudinal serology follow-up in infants. Related issues also raised in question #1C and #20

Figure 4 shows IgG levels from blood samples collected in 48 infants at various time points. We had designed to collect infant blood samples in a more structured manner - at two weeks, two months, and six months during their routine pediatric visits. During the pandemic the visit schedules varied significantly due to parental hesitance to come to the clinics for concerns of COVID exposure. Thus, infants' blood samples were collected anytime between 1-4 weeks, 5-12 weeks, and 13-28 weeks at the time of clinic visits.

We have included this in the Method section and added this explanation in the discussion section.

VERSION 2 – REVIEW

REVIEWER	O'Ryan, Miguel University of Chile
REVIEW RETURNED	09-Jun-2021
GENERAL COMMENTS	Authors have addressed all comments and suggestions and the manuscript is in my perception now suitable for publication.

=